# Significance of Preexisting Vector Immunity and Activation of Innate Responses for Adenoviral Vector-Based Therapy

**DOI:** 10.3390/v14122727

**Published:** 2022-12-06

**Authors:** Wen-Chien Wang, Ekramy E. Sayedahmed, Suresh K. Mittal

**Affiliations:** Department of Comparative Pathobiology, Purdue Institute of Immunology, Inflammation and Infectious Disease, Purdue University Center for Cancer Research, College of Veterinary Medicine, Purdue University, West Lafayette, IN 47907, USA

**Keywords:** adenoviral vector immunity, preexisting immunity, adenoviral immunity, circumvention of preexisting vector immunity, adenovirus capsid, adenovirus tropism, adenoviral innate immunity, hepatic toxicity, blood factor, adenoviral vector vaccine, adenoviral gene therapy

## Abstract

An adenoviral (AdV)-based vector system is a promising platform for vaccine development and gene therapy applications. Administration of an AdV vector elicits robust innate immunity, leading to the development of humoral and cellular immune responses against the vector and the transgene antigen, if applicable. The use of high doses (10^11^–10^13^ virus particles) of an AdV vector, especially for gene therapy applications, could lead to vector toxicity due to excessive levels of innate immune responses, vector interactions with blood factors, or high levels of vector transduction in the liver and spleen. Additionally, the high prevalence of AdV infections in humans or the first inoculation with the AdV vector result in the development of vector-specific immune responses, popularly known as preexisting vector immunity. It significantly reduces the vector efficiency following the use of an AdV vector that is prone to preexisting vector immunity. Several approaches have been developed to overcome this problem. The utilization of rare human AdV types or nonhuman AdVs is the primary strategy to evade preexisting vector immunity. The use of heterologous viral vectors, capsid modification, and vector encapsulation are alternative methods to evade vector immunity. The vectors can be optimized for clinical applications with comprehensive knowledge of AdV vector immunity, toxicity, and circumvention strategies.

## 1. Introduction

Adenovirus (AdV) was first isolated in 1953 from human adenoids while searching for the etiologic agent of acute respiratory infections [1,2], and it was later characterized as human Ad (HAdV) type 5 (HAd5). Currently, more than 100 genotypes of HAdVs are known [3,4]. Overall, AdV-based vectors can infect several types of rapidly dividing or quiescent cells [5]. They can easily be propagated to high titers and purified in large quantities and support the high-level expression of a foreign gene. Activation of innate immunity following AdV vector inoculation leads to antigen-specific humoral and cell-mediated immune responses. Therefore, AdV-based vectors provide an attractive platform for recombinant vaccines and gene therapy.

Vaccines play a vital role in public health by significantly protecting humans and animals from infectious diseases. Most of the currently licensed human vaccines are conventional vaccines in the form of live-attenuated, inactivated, or subunit vaccines. During the COVID-19 pandemic, second-generation vaccine platforms such as AdV vector-based and mRNA-based vaccines were developed [6,7]. An AdV vector for gene therapy can play different roles depending on the need, including the somatic cell expression of a defective gene for correcting metabolic or neurologic disorders [8,9,10,11,12], the use of a tumor suppressor or apoptosis gene [12,13,14], immunomodulation of the host’s immune system [15], induction of immune responses against cancer antigens [16], and selective cell lysis with a conditional replication-competent vector [17]. Invariably, the vector dose for a gene therapy application is many-fold higher than the vector for a vaccine. Therefore, systemic delivery of high doses of AdV vectors for gene therapy can result in vector toxicity in patients [18]. 

The high prevalence of AdV infections in humans results in preexisting AdV immunity, which hinders the efficacy of many HAdV vectors [19]. Moreover, inoculation with an AdV vector will also elicit anti-vector immunity, which may significantly inhibit its effectiveness following the repeat administration of the same vector [20].

## 2. AdV Biology

AdVs are approximately 90–100 nm in diameter and are nonenveloped icosahedral viruses belonging to the *Adenoviridae* family [21], consisting of double-stranded linearized DNA genomes in a range of 25–48 kilobase pairs (Figure 1A). The genome has inverted terminal repeat (ITR) sequences in the range of 26–721 bp [3], depending on the AdV type and the species of origin, at the left and right ends of the genomes, which function as the origins of DNA replication [22]. Next to the left ITR, the genome packaging signal is essential for packaging the AdV genome [23]. The early regions (E1, E2, E3, and E4) encode proteins critical for activating the transcription of other viral genes, preparing the host cell for virus replication, or modulating host immune responses [24,25,26]. The late regions (L1-L5) encode most structural proteins to produce progeny virions [22,24,25].

The hexon (protein II), penton base (protein III), and fiber (protein IV) are the major capsid proteins that create the capsid shell [27]. The 240 hexons form the facet of the icosahedron (12 per facet) and 12 penton bases lie at each vertex of the icosahedron with extensions of 12 fibers [28]. The fiber protein comprises knob and stem domains, and the fiber length varies among AdV types [29]. All major capsid proteins are the targets for neutralizing antibodies; however, the hexon serves as the dominant immunogen for type-specific neutralizing antibodies [30,31]. Four minor proteins (protein IIIa, VI, VIII, IX), known as cementing proteins, help to assemble and support the AdV virion structure at both the inner and outer surfaces [32,33]. The VI and VI precursor proteins (pVI) play critical roles in the endosome escape of AdV after endocytosis and activate the cleavage of several precursor proteins [34]. Moreover, six core proteins (V, VII, IVa2, terminal protein, Mu, and protease) are responsible for DNA replication and transcription, DNA condensation, and viral assembly [33,35]. A schematic virion structure of HAdV5 consisting of at least 13 proteins is presented (Figure 1B). According to the phylogenetic classification, genome organization, and biological features, AdVs are classified into six genera: *Mastadenovirus*, *Aviadenovirus*, *Atadenovirus*, *Siadenovirus*, *Ichtadenovirus*, and *Testadenovirus* [3]. Furthermore, based on their hemagglutination properties, sequence correlation, and oncogenic characteristics, HAdVs are categorized into seven species (A-G) [3,36,37].

## 3. AdV Receptors and Cell Entry

AdVs utilize the distal knob domain to bind to their respective primary cell receptors with high affinity and initiate endocytosis (Figure 2) [38,39]. Many HAdVs recognize the Coxsackie and AdV Receptor (CAR) on the cell surface for attachment and internalization [40]. CAR is a transmembrane glycoprotein consisting of two immunoglobulin-like extracellular domains, a transmembrane domain, and a long cytoplasmic domain. The immunoglobulin-like domain (D1) seems necessary for viral binding [41]. CAR is a component at the tight junction of epithelial cells mediating cell–cell adhesion and is broadly expressed in different tissues [40,42]. CAR expression on non-permissive cells increases the repertoire of AdV-mediated gene transfer [43,44]. Meanwhile, type B HAdVs utilize CD46, CD80/86, or receptor X as the cell receptor for attachment [45,46,47,48,49,50]. Moreover, some AdVs use receptors other than CAR for cell entry, such as desmoglein-2 for HAdV3, 7, 11, 14 [51], glycans GD1a for HAdV37 [52], and sialic acid for HAdV37 and 52 [53,54]. AdVs of different species, such as bovine, porcine, or ovine, use CAR-independent mechanisms for cell attachment and internalization [55,56,57,58].

Interaction between CAR and AdV fiber knobs displaces the fibers, thereby exposing the penton base [42,59]. It allows the attachment of the RGD domain of a penton with αv integrins, a heterodimeric transmembrane protein involved in cell adhesion, and stimulates virus internalization by facilitating clathrin-coated endocytosis [60,61]. Subsequently, the activation of several GTPases, such as dynamin, PI3K, Rac1, Cdc42, and Rab5, results in virus internalization into the clathrin-coated endosome [39,62,63]. The acidic environment in the endosome results in the dissociation of the penton base and peripentonal hexon from the virion, followed by protein VI-mediated endosomal lysis [61,64]. The microtubule motors transport the uncoated viral particle to the nuclear membrane. Subsequently, the viral genome is injected into the nucleus via the nuclear pore complex [65,66].

## 4. AdV Vector Construction

AdVs’ genome manipulation for vector generation for vaccine development and gene therapy applications has evolved over the years [14,67]. Initially, the replication-competent AdV vector was generated by deleting the E3 region, which is not essential for virus replication [68]. The deletion of E1 and E3 was introduced into the AdV vector to increase the size of the foreign gene cassette, and this type of replication-defective vector is considered the first-generation AdV vector [69]. The E1- and E3-deleted vectors can easily be propagated in cell lines that constitutively express E1 proteins, such as HEK 293 [70] and PER.C6 [71]. Several techniques have been developed to construct AdV vectors. The investigators have used homologous recombination in bacteria [72], the Cre-lox recombination system [73], and homologous recombination in the appropriate cell lines [74] to generate AdV vectors. Due to the leaky expression of some of the AdV proteins, the development of AdV-specific immune responses at high levels will eliminate the transduced cells, thereby shortening the duration of transgene expression [75]. Therefore, the second generation of AdV vectors was developed by extending deletions in the E2 and/or E4 regions, in addition to E1 and E3 deletions. These vectors have a foreign gene insertion capacity of up to 14 kb, with improved vector survival in the host [76,77].

Nevertheless, the vector yield per cell is less than in the first-generation vectors because of the reduced replication in the producer cell lines [67]. The third-generation AdV vectors, known as helper-dependent or gutless AdV vectors, were generated by the deletion of the viral genome except for the ITRs and packaging signal. A particular AdV helper virus is used for generating helper-dependent AdV vectors having a larger insert capacity with low toxicity [78,79,80]. Moreover, the conditional replication-competent AdV vectors or oncolytic AdV vectors can be generated by controlling the E1 expression with an inducible promoter [17]. The first-generation vectors are preferred for vaccine development, while the second, third, and conditional replication-competent AdV vectors are better suited for gene therapy. The implications of oncolytic AdV vectors have been reviewed in detail elsewhere [81,82]. 

Several unique characteristics of AdV vectors have contributed to their potential as vectors for gene delivery. The AdV genome can easily be manipulated, allowing the insertion of a sizeable foreign gene cassette [67], which remains stable even after multiple passages, and the vector system has an excellent safety record [83,84]. AdV vectors contain various pathogen-associated molecular patterns (PAMPs) interacting with pathogen recognition receptors (PRR), thereby stimulating robust innate immune responses leading to enhanced adaptive immunity [36,85,86,87]. Moreover, AdV vectors can quickly be grown and purified in large quantities and have broad host cell tropism [5,88]. Furthermore, AdV vector vaccines can be administrated through the mucosal route, such as the oral and the nasal routes [89]. AdV vector vaccines administered through the mucosal surfaces can induce cellular and mucosal immunity [89]. Mucosal immunity can prevent the initial entrance of pathogens, which is a great advantage against respiratory pathogens [90,91].

## 5. AdV-Mediated Activation of Innate Immunity

The host immune system can recognize PAMPs in microbial structures or products through PRRs on the cell surface, resulting in an innate immunity cascade [87,92]. One of the best-characterized types of PRRs is Toll-like receptors (TLRs), which are expressed on antigen-presenting cells (dendritic cells and macrophages) as well as non-immune cells such as fibroblasts and epithelial cells [93]. The presence of various PAMPs in AdVs results in the activation of multiple innate immune pathways [36,86]. The innate immunity is initiated by the sensing of AdV through interactions between viral proteins and cellular receptors. The binding of viral fiber to CAR (with many AdVs) and/or interactions between the penton base and integrins induce the downstream signaling of nuclear factor kappa B (NF-κB), resulting in the increased expression of proinflammatory cytokines and chemokines [94,95,96]. In addition, the binding of the blood coagulation factor X (FX) to the AdV hexon is associated with the activation of TLR4 adaptor molecules such as differentiation primary response 88 (MyD88), TNF receptor-associated factor 6 (TRAF6), and TIR domain-containing adaptor-inducing interferon-β (TRIF), following the stimulation of TLR4 [97]. Upregulation of NF-κB results in the enhanced expression of several proinflammatory cytokines and chemokines, such as IL1α, IL1β, IFNα, TNFα, IL2, IL6, IL8, CCL2, CCL3, CCL4, CXCL1, CXCL2, CXCL10, GM-CSF, and others [97,98,99]. The levels of cytokines’ and chemokines’ expression are dependent on the AdV vector dose and the host. In a mouse model, the elevated cytokines’ and chemokines’ levels are detected within 0.5–3 h, reaching the peak at 6–12 h and then declining to normal levels at 7–10 days post-inoculation with an AdV vector [98]. Following the intravenous inoculation of an AdV vector, tissue-resident macrophages, including Kupffer cells in the liver and CD169^+^ and MARCO^+^ macrophages in the spleen, secrete the proinflammatory cytokine IL1α within 10 min [94]. The activation of IL1α triggers the expression of other proinflammatory cytokines, including IL6 and TNFα, and chemokines CCL2, CXCL1, and CXCL2, within 30 min [100]. 

The AdV genome can be detected in the endosomes by TLR9, an endosomal sensor that recognizes the double-stranded DNA containing unmethylated CpG motifs and initiates cytokine secretion [101,102,103,104]. Moreover, the stimulation of innate immunity can also be mediated by TLR-independent mechanisms through the DNA-sensing pathway, which induces interferon (IFN) regulatory factor 3 (IRF3) and the NLR family pyrin domain containing 3 (NLRP3) inflammasome, thereby initiating IFN type I production [87,105,106]. Furthermore, the induction of IRF7 occurs following vector escape from endosomes stimulating IFNα/β production in splenic myeloid dendritic cells [107]. The expression of type I IFN is detected within 4 to 6 h after the intravenous administration of HAdV2 [100]. Moreover, the intravenous administration of HAdV2 or HAdV5 results in the initiation of complement pathways and a rapid increase in the complement fragment C3a within 10 min, reaching the peak at 30 min [108]. Complement activation by HAdV2 or HAdV5 does not need antibodies in vivo and they can be activated through both classical and alternative pathways [108]. The natural route of AdV infection may induce a low level of inflammatory response. In contrast, the intravenous administration of a higher number of AdV particles results in robust innate immunity with a high level of several cytokines and chemokines [98]. It may lead to cytokine storm syndrome, hepatotoxicity, disseminated intravascular coagulates, and thrombocytopenia, leading to adverse effects and even death [98,100]. The negative reaction occurs with significantly high vector doses, especially for gene therapy applications [109]. The considerably high amount of vector accumulation predominately in the liver and spleen will initiate an acute inflammatory response due to the induction of high levels of chemokines and cytokines, followed by the recruitment of neutrophils [110,111]. Systemic vector toxicity in the form of significant hepatic necrosis and apoptosis was observed in mice following the intravenous administration of the HAdV5 vector at a high dose (1 × 10^10^ PFU) [111]. Similarly, tissue damage following the intraportal administration of a nonhuman AdV vector was also noticed in primates [112]. Moreover, the induction of type I IFN following the inoculation of an AdV vector could restrict the duration of transgene expression due to the activation of natural killer (NK) cells [113]. The activated NK cells participate in the rapid elimination of AdV vector-transduced cells, thereby shortening the duration of transgene expression [113,114]. Induction levels of inflammatory cytokines and chemokines may vary with the type of AdV vector, resulting in variability in transgene-specific immunogenicity [115]. For AdV vector-based vaccines, the quality of transgene-specific immune responses is more critical than the duration of transgene expression. However, the level and duration of transgene expression and vector survival are vital for many gene therapy applications. 

## 6. AdV Vector-Mediated Adaptive Immunity

AdV vector-induced innate immunity triggers adaptive immune responses against the vector and the transgene. Following AdV vector inoculation, the APCs transport the vector, vector proteins, or expressed transgene to the draining lymph nodes, where the naïve T cells are primed and differentiate into CD4^+^ and CD8^+^ T cells [116]. With the recognition of the viral or transgene epitopes with the MHC class I or II molecules, cellular and humoral immunity is elicited; after this, vector-specific neutralizing antibodies and CD8+ T cells eliminate the vector and vector-infected cells [116,117]. Nevertheless, vector-specific adaptive immunity significantly affects the transgene expression while removing the vector-infected cells, thus posing a challenge to the AdV vector-based gene therapy [118].

The development of anti-AdV neutralizing antibodies (NAbs) contributes towards the inhibition of the vector’s efficacy. The adoptive transfer of HAdV5-specific NAbs to mice negatively impacted the development of transgene-specific immune responses after the administration of an HAdV5 vector-based vaccine [119]. Meanwhile, the depletion of antibodies against the hexon, penton, and fiber significantly improved the transduction efficiency of an AdV vector [120]. The AdV major capsid proteins—hexon, penton, and fiber—are the targets for NAbs against the vector, while the hexon has the predominant reactive sites for NAbs [121]. In human blood samples from the U.S. and developing countries, hexon-specific NAbs titers against HAdV5 were significantly higher than fiber-specific NAbs [122]. The hexon-specific NAbs showed better inhibition of an HAdV5 vector-based vaccine than the fiber-specific NAbs [122]. There are seven hypervariable regions (HVRs) in the flexible loops located at the outer surface of the hexon [123]. HVRs are diverse among AdVs at the amino acid level and contain critical virus-neutralizing epitopes [124]. Anti-fiber Abs can block primary cell receptor interactions, and anti-penton Abs can hinder integrin-mediated internalization [125]. Moreover, the anti-penton Abs showed a synergistic effect with anti-fiber Abs on AdV neutralizing activity and reduced vector-mediated transduction [126,127]. Natural infections with HAdV5 elicit a higher level of fiber-specific NAbs than vaccination with an HAdV5-based vaccine [128]. Different circumstances may present distinct vector-neutralization mechanisms, while the hexon and fiber play vital roles in vector neutralization [121,128]. Therefore, hexon or fiber modifications can circumvent AdV NAbs and serve as one of the strategies to enhance vector efficacy [121,129]. Antibody-mediated AdV neutralization occurs due to virus aggregation, blocking AdV attachment and entry into the host cells [130]. Other mechanisms, such as AdV particle destabilization, abortive virus internalization, and interference of virus uncoating, are also induced by NAbs [125,131,132]. Unique post-entry AdV neutralization by blocking the microtubule-dependent virion translocation with hexon-specific NAbs was also elucidated [125,133]. 

In addition to AdV NAbs, cell-mediated immunity plays a critical role in AdV clearance [134]. AdV-specific NAbs play a dominant role in virus clearance compared to AdV-specific CD8+ T cells [111]. The AdV-specific CD4^+^ and CD8^+^ T cells were detected in healthy adults in peripheral blood mononuclear cells (PBMC) [135,136]. Unlike NAbs, AdV-specific T cell responses are cross-reactive among different AdV types [137]. A CD4 T cell epitope on hexon, H910–924, was identified as a conserved epitope among diverse AdV types [137]. This conserved epitope induces robust T cell proliferation and is recognized by 78% of healthy adults [137,138]. Moreover, four conserved CD8^+^ T cell epitopes on the hexon can also induce cross-reactive T cell immunity [139].

The durability of humoral and cell-mediated immune responses is critical for the efficacy of AdV vector-based gene delivery [140]. AdV-specific memory CD8 T cells are vital for long-term cell-mediated immunity, which helps in the rapid expansion of these cells in response to the production of antiviral cytokines and cytotoxic molecules following AdV re-infection or vector inoculation [141]. Only a small proportion, around 5–10%, of the activated T cells will become long-lived memory T cells [142]. AdV-specific CD4^+^ and CD8^+^ memory T cells are detected in healthy humans from a natural AdV infection [143]. Most AdV-specific CD4^+^ T cells displayed a central memory-like phenotype, whereas CD8^+^ T cells exhibited an effector-like phenotype [143]. For AdV vector-based vaccine applications, the immunogen-specific memory T cells were detected in vitro and in vivo [144,145]. These memory T cells perform multiple functions, including degranulation and the production of cytokines [144]. Interestingly, less prevalent HAdVs, such as HAdV26, HAdV35, and HAdV48, showed better CD8^+^ T cell memory than HAdV5. Enhanced expansion of CD8^+^ T cells was elicited by using two AdV types for the prime–boost approach [145]. Using more than one AdV vector type will confer better outcomes for vaccine and gene therapy applications compared to multiple doses with the same vector.

## 7. AdV Tropism, Blood Factors, and Vector Toxicity

While CAR is widely distributed to multiple cell types, the systemic administration of an AdV vector results in predominant tropism to the liver [146]. A biodistribution study in mice inoculated intravenously with an AdV vector demonstrated that the highest amount of transgene expression was detected in the liver [147]. This restricted tropism limits vector distribution to other tissues and leads to liver damage and systemic toxicity [148,149]. The ablation of CAR-binding by modifying the AdV vector did not significantly inhibit biodistribution and hepatotoxicity, implying that other factors besides CAR interaction are vital for liver tropism and hepatotoxicity [150].

Interactions between an AdV vector and the host’s blood factors play a significant role in AdV liver transduction [151]. The modification of the CAR-binding domain on the AdV fiber inhibits the infection of hepatocytes in vitro, but the transfection efficacy of hepatocytes remains unaffected in vivo [151]. This in vivo liver tropism is due to the binding of the blood coagulation factor (F) IX (FIX) and complement factor C4BP to the AdV vector, resulting in interactions with the hepatocellular receptors, including heparan sulfate proteoglycan (HSPG) and low-density lipoprotein receptor-related protein (LRP) for liver transduction [151,152]. The vitamin K-dependent coagulation factors FX, FVII, and protein C can also enhance the hepatocyte transduction of AdVs [153]. In addition, the AdV uptake by Kupffer cells is mediated via the blood factors [151], initiating the production of proinflammatory cytokines such as TNF and IL6, followed by liver damage after the systemic administration of 1 X 10^10^ transduction units of HAdV5 vector expressing human α1-antitrypsin [154]. The γ-carboxyl glutamic acid (GLA) domain is common among vitamin K-dependent coagulation factors [155]. Unlike the fiber interaction of FIX, the GLA domain of FX can attach to the HVR5 and HVR7 of the hexon in the presence of calcium ions, resulting in efficient transduction of the hepatocytes [97,156,157]. Apart from FX, other coagulation factors showed weak affinity toward the hexon of HAdV5 [109]. This implied that these factors might affect vector tropism through different mechanisms [109,156]. 

Interactions with coagulation factors may also vary with the AdV type. HAdV31, belonging to species A, interacts with FIX but not with HAdV5, a species C AdV [158]. In comparison, a species D AdV showed weak FX-binding affinity, leading to low hepatic tropism due to a distinct hexon HVR [156]. Understanding receptor binding, coagulation factors’ interactions with AdV, the availability of AdV types, and AdV capsid modifications could help in AdV vector design to lower hepatic tropism and vector toxicity. The fiber shortening or fiber exchange from another AdV type can alter the tropism and significantly decrease the hepatic transduction and toxicity [148]. Mice immunized with an AdV vector, having poly-lysine in the fiber’s knob domain, showed lower levels of IL6 and aspartate aminotransferase (AST), indicating that vector attenuation led to decreased liver toxicity [159]. Replacing the hexon HVR of an AdV vector from a non-FX-binding AdV type can also abolish the coagulation factor-mediated liver transduction [157]. Moreover, nanoparticle coating or polymer encapsulation of AdV vectors effectively decreased liver tropism [160]. Furthermore, advanced AdV retargeting is required to target specific cell types, especially for cancer therapy. A tissue-specific promoter in the E1 region to develop conditional replication-competent AdV vectors, tissue-specific ligand insertion in the fiber knob, and bifunctional adapter molecules are some strategies for targeting specific tissues with AdV vectors [160,161,162].

## 8. Preexisting Ad Vector Immunity and Its Implications

There is a high prevalence of HAdV infection due to exposure to one or more HAdV types since childhood, due to the availability of over 85 types of HAdVs [163]. Serosurveillance studies showed that 40–90% of humans have NAbs against HAdV5 [163], the most studied HAdV. It seems that children and young adults usually have higher levels of NAbs against AdV than the elderly [19,164]. The development of AdV-specific NAbs following natural infection with one or more AdV types is known as preexisting AdV immunity or preexisting AdV vector immunity, which affects the efficacy of HAdV vectors (Figure 3) [115]. A clinical trial with an HAdV5-based Ebola vaccine showed that volunteers with preexisting AdV vector immunity had a lower response to the vaccine than participants without preexisting AdV immunity [165]. In a clinical trial with the HAdV5-based SARS-CoV-2 vaccine, vaccinees having high preexisting anti-HAdV5 immunity showed lower levels of humoral immune responses compared to participants with low levels of preexisting anti-HAdV5 immunity [166]. Preexisting AdV immunity negatively impacts the vector uptake and development of immunogen-specific adaptive immune responses, thereby affecting the Ad vector’s efficacy [167]. Furthermore, residents of sub-Saharan Africa showed preexisting neutralizing antibodies against some chimpanzee AdVs (ChAdVs), which may restrict the application of ChAdV as an alternative vaccine vector [168].

In addition to preexisting AdV immunity, the first inoculation with an AdV vector will also lead to vector-specific immune responses, impacting the subsequent administration of the same vector [20]. To evaluate the durability of vector immunity and its effect on the development of immunogen-specific immune responses, naïve or HAdV5-primed mice were immunized with the HAdV-H5HA (HAdV5 vector expressing H5N1 hemagglutinin (HA)) at 1, 3, 6, and 10 months after priming and challenged with an antigenically distinct H5N1 influenza virus four weeks following vaccination [169]. With time, there was a continuous decline in vector immunity, with a constant improvement in HA-specific immune responses and enhanced protection against a heterologous H5N1 virus [169]. The drop in vector immunity at six months and onwards was sufficient for inducing the required level of protective immunity. This study implies that vector immunity can inhibit vector efficacy following repeated vaccination with the same vector. Moreover, annual immunization with the same vector should be possible due to the decline in vector immunity to levels that may not impact the vector’s efficacy significantly. The prime–boost approach, using different AdV type vectors, is an excellent strategy to substantially improve the effectiveness of the AdV vector platform by eluding the vector-specific immunity during the boosting step [170]. This approach was used for the HAdV5/HAdV26-based prime–boost vaccine for SARS-CoV-2 [171]. An HAdV11 vector, which showed better transduction efficiency in smooth muscle cells, synoviocytes, dendritic cells, and cardiovascular tissues compared to HAdV5, was not hampered by preexisting HAdV5-specific NAbs and was investigated as a vector for gene therapy [74].

## 9. Strategies to Circumvent Preexisting Vector Immunity

HAdVs with low seroprevalence in humans, including HAdV11, 35, or 50 of species B, and HAdV26, 48, or 49 of species D, are unaffected by preexisting AdV immunity [172,173,174]. In particular, the HAdV35 and HAdV26 vectors were evaluated in humans for their vaccine efficacy against *Mycobacterium tuberculosis*, malaria, HIV, Ebola virus, and SARS-CoV-2 [175,176,177,178,179,180]. For gene therapy, a HAdV35 vector demonstrated better transduction capability towards glioma cells, indicating its utility in treating malignant glioma [181]. However, six rare HAdVs (HAdV11, HAdV26, HAdV35, HAdV48, HAdV49, and HAdV50) showed lower immunogenicity than the HAdV5 vector [174]. HAdV35- or HAdV26-based Ebola vaccines induced lower immunogenicity and protection against the Ebola virus in nonhuman primates compared to the HAdV5-based vaccine [115]. Since the AdV-specific cellular immunity is comparatively cross-reactive, HAdV26-specific cellular immune responses at a baseline were detected in humans without HAdV26-specific NAbs [182]. The HAdV35-specific CD4^+^ T cells were detected in individuals without HAdV35-specific NAbs [135]. These studies suggest that AdV-specific T lymphocytes are cross-reactive among multiple AdV types. On the other hand, HAdV35, HAdV26, and HAdV48 interact with CD46 as the primary cellular receptor and induce enhanced innate immunity compared to HAdV5 [183]. The rare HAdVs offer attractive alternatives to avoid preexisting vector immunity for efficient transgene expression. Several strategies that are utilized to circumvent the preexisting vector immunity are outlined below (Figure 4).

Alternatively, nonhuman AdV vectors of simian, bovine, porcine, ovine, canine, or avian origin were developed to evade the AdVs’ preexisting immunity since AdVs are species-specific and do not naturally infect humans [184]. Cross-NAbs against ChAdV are rarely detected in developed countries’ populations but showed higher prevalence in sub-Saharan Africa [185]. Replication-defective ChAdV vectors can grow efficiently in the cell lines that support the replication of HAdV5 vectors, e.g., the HEK 293 cell line [186]. In addition, ChAd vectors can elicit high levels of humoral and cellular immune responses against the immunogen, similar to the HAdV5 vector expressing the same immunogen, and the immunogenicity is not significantly inhibited in the presence of preexisting HAdV immunity [187,188]. However, there is variability in the resultant immunogenicity with different ChAdV types [189]. ChAdV vector vaccines against SARS-CoV-2, Ebola virus, malaria, HIV, and respiratory syncytial virus have been evaluated for their efficacy in humans [190,191,192,193]. AstraZeneca’s SARS-CoV-2 vaccine based on the ChAdV vector platform (ChAdOX1) was licensed in several countries [180]. The bovine Ad (BAdV) type 3 (BAdV3) belonging to the *Mastadenovirus* genus was generated as a gene delivery vector [194] and showed excellent potential in overcoming preexisting vector immunity [195]. Preexisting HAdV NAbs do not cross-react with BAdV3 or hamper the transduction ability of the BAdV3 vector [195]. The HAdV-specific cellular immunity showed minimal cross-reactivity with the BAdV3 vector [196]. Moreover, the BAdV3 vector can efficiently transduce various tissues, with relatively prolonged persistence compared to the HAdV5 vector [197]. Better stimulation of innate immunity with higher expression of proinflammatory chemokines and cytokines was also shown in mice inoculated with a BAdV vector compared to that of a HAdV5 vector [198]. A BAdV3 vector-based influenza vaccine provided better immunogenicity and protection than the HAdV5 vector-based influenza vaccine [199]. Since BAdV3 utilizes sialic acid as the receptor for virus entry [55], this vector platform is suitable for nasal delivery [200]. It seems that there is enhanced antigen processing through the autophagy pathway in mice immunized with a BAdV3 vector-based vaccine compared to the HAdV5 vector-based vaccine [200]. Overall, the BAdV3 vectors circumvent the preexisting vector immunity and confer robust immunogenicity. Other nonhuman AdV vectors, such as porcine AdV, ovine AdV, canine AdV, and avian AdV, are also under evaluation as alternative vectors for gene delivery [187].

Since the capsid proteins are the main targets of NAbs, modification of the capsid proteins is a relevant strategy to overcome preexisting vector immunity. The HVR sequence of the HAdV5 hexon was exchanged with the corresponding regions from HAdV48 to design a hexon-chimeric AdV vector [201]. This vector showed robust immunogenicity in the presence of high levels of preexisting anti-HAdV5 NAbs [201]. Similarly, fiber-chimeric AdV vectors were generated by exchanging the knob domain of HAdV5 with the knob region of HAdV3 [129]. This also enhanced its ability to circumvent preexisting HAdV5 immunity in mice pre-immunized with the unmodified HAdV5 vector [129]. There was increased transduction efficiency of a chimeric HAdV5 vector with the HAdV35 fiber in HAdV5-immunized mice [202]. The same strategy can also be extended to nonhuman AdVs. A chimeric HAdV5 vector with the BAdV4 fiber demonstrated reduced innate immunity and interactions with blood coagulation factors in mice compared with the HAdV5 vector [203]. These modifications were also used for gene therapy applications to redirect the AdV tropism to the targeted cells [204,205].

Chemical encapsulation can shield the antigenic epitopes on the AdV surface and thus help to evade the AdV NAbs. PEGylation of the AdV vector can retain the virus transduction capability but protect the vector from NAbs [206]. The transgene expression with the PEG-conjugated AdV vector was higher than that with the untreated vector in mice preimmunized with the unmodified AdV vector [207]. Moreover, the PEGylation of AdV vectors reduces the levels of vector-induced innate immunity and thereby decreases vector toxicity, such as thrombocytopenia or increased levels of ALT [207]. PEGylation with PLGA encapsulation also enhanced vector stability and gene transfection efficiency in vitro [208]. Moreover, encapsulation of AdV vectors with liposomes can also provide resistance to AdV NAbs [209], enhancing the transduction efficiency with repeated vector administration [210]. It can also reduce hepatic uptake and improve the transduction of CAR-deficient cells [211,212]. PEGylation in liposome-encapsulated AdV vectors can further reduce cytotoxicity, hemolytic activity, anti-vector immunity, and innate immunity [213]. Moreover, the microencapsulation of AdVs into biodegradable sodium alginate microspheres also eluded vector-specific immunity [20].

Sequential inoculations with different routes are an alternative strategy to conquer the anti-vector immunity during repeat administration. It was demonstrated that the preexisting immunity from the intramuscular injection of an AdV vector does not significantly affect the immunogenicity of the same AdV-based vaccine when administrated intranasally [214]. Similarly, mice pre-immunized with HAdV5 intramuscularly or intranasally do not show decreased humoral immunity to the transgene from subsequent HAdV vector oral administration [215]. The intranasal administration of AdVs infects localized antigen-presenting cells that carry the antigen into the local draining lymph node. This causes most activated effector T cells to be localized in the respiratory system, and long-living memory T cells remain retained within these tissues [91]. Parenteral administration usually circulates the effector T cells into systemic lymphoid organs, with minimal memory T cells in the respiratory tract [91]. 

## 10. Conclusions and Future Directions

AdV vectors have been at the forefront of promising gene delivery platforms for vaccines and gene therapy. The emergency-use authorization of three AdV vector-based SARS-CoV-2 vaccines is one of the AdV vector platform’s success stories. This platform has extensively been utilized as a gene delivery system for vaccine or gene therapy applications in several human clinical trials. With our advancing understanding of AdV biology, vector design has versatility depending on the need. However, the preexisting vector immunity and induction of enhanced innate immunity at higher vector doses are notable limitations that could significantly affect the efficacy of the AdV vector platform. The development of rare human and nonhuman AdVs as gene delivery platforms has helped to overcome the preexisting vector immunity issue. The concept of a prime–boost approach with two different AdV vector types, or priming with an AdV vector followed by a boost with another vector system, could significantly reduce the adverse impact of innate immunity. 

Additional work is needed to design strategies for further controlling vector tropism to enhance its effectiveness while significantly inhibiting hepatic toxicity, especially for gene therapy applications. Activation of innate immunity by AdV vectors is a unique strength, vital for the success of AdV vector-based vaccines, by eliciting enhanced immunogen-specific immune responses. Therefore, the induction of innate immunity needs to be better managed to avoid adverse effects without compromising its adjuvant impact. AdV vectors for mucosal immunization require further exploration to provide better vaccine efficacy against mucosal pathogens. Investigation of other rare human or nonhuman AdVs for vector development should continue to expand the vector choices for various gene therapy applications.

Further investigations to determine strategies to inhibit the AdV vector binding to the blood factors will be critical for improving the vector’s versatility. The worth of the same or different AdV vector for annual vaccination needs to be explored. New research in nanoparticles and AdV capsid modifications will further boost the quality and efficacy of AdV vector-based gene delivery applications. Additional work is necessary to further enhance the durability of immunogen-specific immunity and the duration of transgene expression when delivered through an AdV vector. 

## Figures and Tables

**Figure 1 viruses-14-02727-f001:**
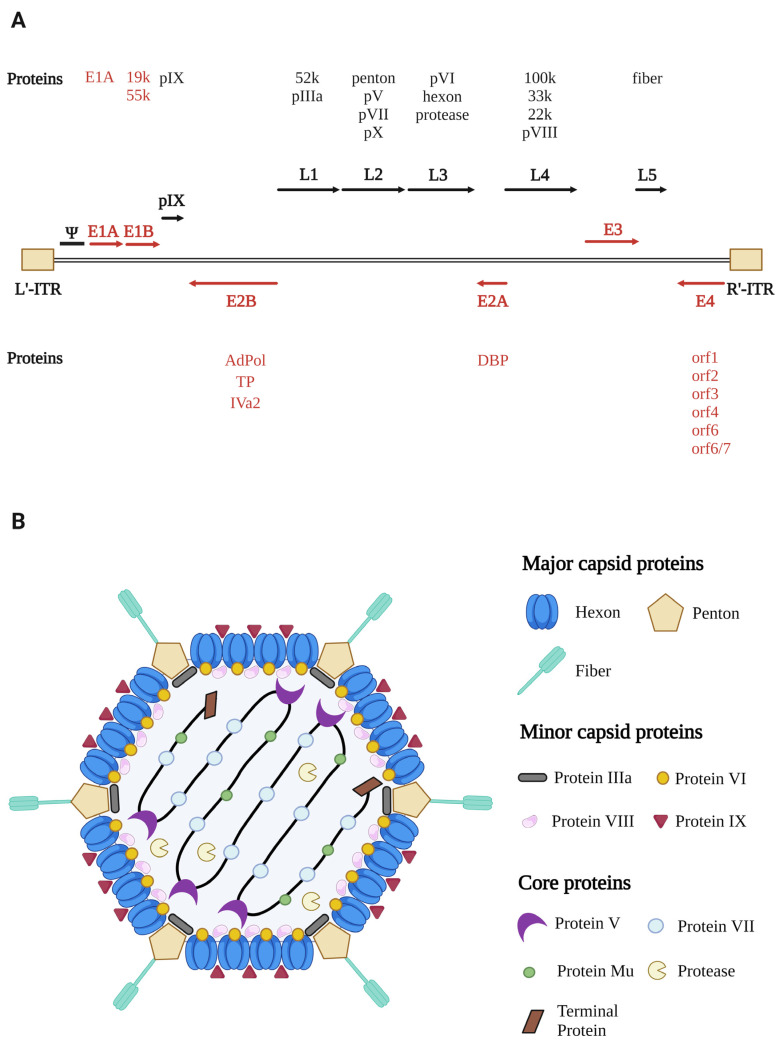
Genome organization and structure of human adenovirus type 5. (**A**) Schematic representation of the adenoviral genome, transcription units, and major proteins. (**B**) Structural representation of adenovirus and its components. E, early region (shown as red arrows); L, late region (shown as black arrows); L’-ITR, left inverted terminal repeat; R’-ITR, right inverted terminal repeat; Ψ, packaging signal; AdPol, adenovirus DNA polymerase; TP, terminal protein; DBP, DNA-binding protein.

**Figure 2 viruses-14-02727-f002:**
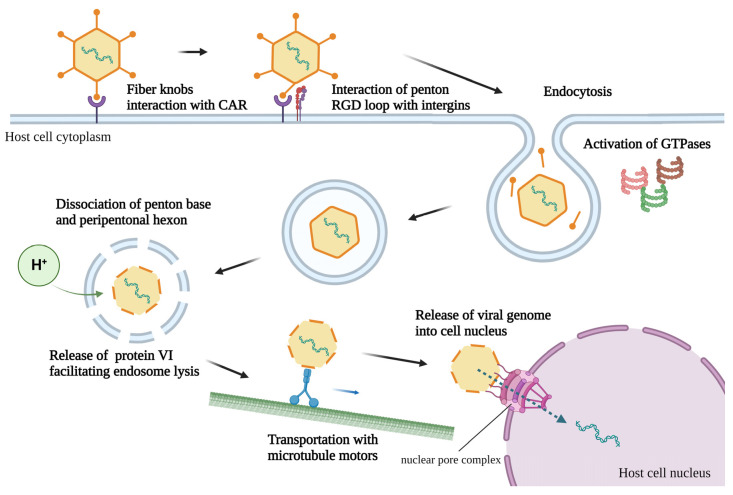
Transportation of human adenovirus type 5 or its genome within a host cell. The adenoviral knob domain binds to the CAR on the host cell surface. Following receptor binding, the fibers start to disassociate, exposing the RGD loop of the penton to interact with αv integrins, thereby initiating endocytosis. The endosome’s acidic environment results in the dissociation of the penton base and peripentonal hexon, followed by protein VI-mediated lysis of the endosome. The viral particle is transported to the nuclear membrane by microtubule motors. Finally, the adenoviral genome is transported to the nucleus through the nuclear pore complex. CAR, Coxsackievirus and adenovirus receptor; RGD, arginine–glycine–aspartic acid; GTPase; guanosine triphosphate hydrolase.

**Figure 3 viruses-14-02727-f003:**
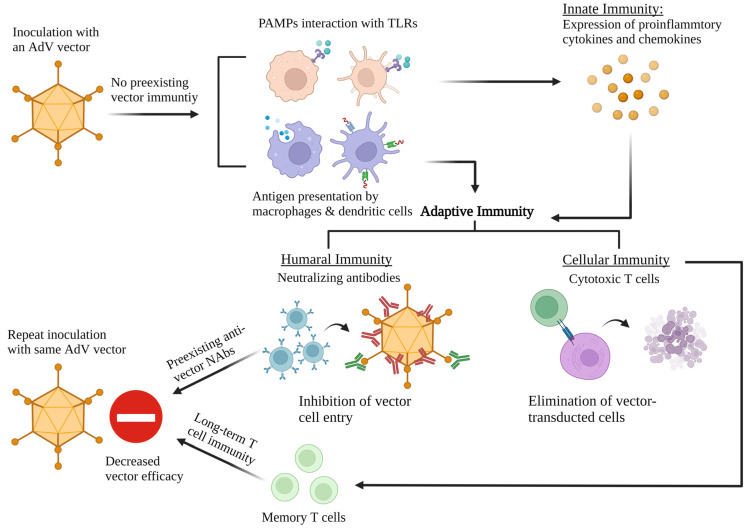
Activation of innate and adaptive immune responses in response to an adenoviral (AdV) vector. Following inoculation with an AdV vector, activation of innate immunity leads to the expression of proinflammatory chemokines and cytokines. Following antigen expression, processing, and presentation, humoral and cell-mediated immune responses develop. The resultant vector immunity will eliminate the vector, leading to a reduced duration of transgene expression. The development of vector-specific neutralizing antibodies (NAbs) and memory T cells provide long-term vector immunity and suppress subsequent inoculation with the same AdV vector. PAMP, pathogen-associated molecular patterns; TLRs, Toll-like receptors.

**Figure 4 viruses-14-02727-f004:**
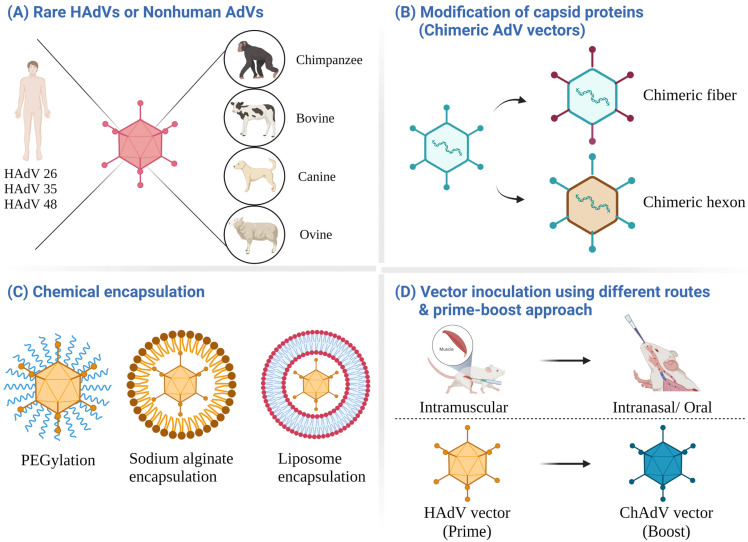
Circumventing strategies against preexisting vector immunity. (**A**) Inoculation with a rare human adenoviral (HAdV) or nonhuman AdV can evade the preexisting vector immunity. (**B**) Since most of the neutralizing antibodies (NAbs) target the capsid proteins, such as the fiber and hexon, exchanging the capsid and/or fiber proteins with a different AdV type can elude the preexisting vector immunity. (**C**) Chemical encapsulation can mask the AdV vector from antibody-mediated neutralization. (**D**) The use of different routes of inoculation or different vector platforms (prime–boost approach) can circumvent the preexisting vector immunity. ChAdV, chimpanzee adenovirus; PEG, polyethylene glycol.

## Data Availability

Not applicable.

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
