# Peer review of "Significance of Preexisting Vector Immunity and Activation of Innate Responses for Adenoviral Vector-Based Therapy"

_viruses, 2022, doi:10.3390/v14122727_

Round 1

Reviewer 1 Report

The review entitled “Significance of preexisting vector immunity and activation of innate responses for adenoviral vector-based therapy” is valuable to AdV vaccine and gene therapy development. I recommend it to be accepted after minor revision.

The official abbreviation for adenovirus is AdV.

Section: 2.Ad biology: The taxonomy of AdV should be included.

Section 3.Ad receptors and cell entry: The receptors are highly associated with the AdV species and types.

Section 4. Ad vector construction: How about the construction of other types of AdVs, e.g., HAdV-11. ["Novel Replication-Incompetent Vector Derived from Adenovirus Type 11 (Ad11) for Vaccination and Gene Therapy: Low Seroprevalence and Non-Cross-Reactivity with Ad5." Journal of Virology.  2004, 78(23): 13207-13215.]. What is the potential for the live AdV vaccine vector? ["Construction and characterization of a replication-competent human adenovirus type 3-based vector as a live-vaccine candidate and a viral delivery vector." Vaccine.  2009, 27(8): 1145-1153.]  AdV vectors used for mucosal immunization are suggested to be discussed in this review.

Section 8. Preexisting Ad vector immunity and its implication.  Simian AdVs have been used as novel viral vectors. However, the preexisting immunity to chimpanzee adenoviruses among African residents seems to have negative effect on SAdV vaccine [Zhang Q, Seto D. 2015. Chimpanzee Adenovirus Vector Ebola Vaccine--Preliminary Report. N Engl J Med 373:775-776.]  The authors should discuss this kind of vectors in this section.

Author Response

Reviewer #1

Comment: The review entitled “Significance of preexisting vector immunity and activation of innate responses for adenoviral vector-based therapy” is valuable to AdV vaccine and gene therapy development. I recommend it to be accepted after minor revision.

Response: We appreciate the reviewer’s valuable time in reviewing the manuscript and your insightful comments.

Comment: The official abbreviation for adenovirus is AdV.

Response: All the abbreviation for adenovirus is replaced by “AdV.”

Comment: Section: 2. Ad biology: The taxonomy of AdV should be included.

Response: We have added information on the taxonomy (section 2; lines 82-85).

Comment: Section 3. Ad receptors and cell entry: The receptors are highly associated with the AdV species and types.

Response: The relevant information has been added (section 3; lines 97-99).

Comment: Section 4. Ad vector construction: How about the construction of other types of AdVs, e.g., HAdV-11. ["Novel Replication-Incompetent Vector Derived from Adenovirus Type 11 (Ad11) for Vaccination and Gene Therapy: Low Seroprevalence and Non-Cross-Reactivity with Ad5." Journal of Virology.  2004, 78(23): 13207-13215.]. What is the potential for the live AdV vaccine vector? ["Construction and characterization of a replication-competent human adenovirus type 3-based vector as a live-vaccine candidate and a viral delivery vector." Vaccine.  2009, 27(8): 1145-1153.]  AdV vectors used for mucosal immunization are suggested to be discussed in this review.

Response: The manuscript is revised to add the suggested information (lines 122-125;

149-153; & the AdV mucosal immunization is discussed in section 9 lines 446-457).

Comment: Section 8. Preexisting Ad vector immunity and its implication.  Simian AdVs have been used as novel viral vectors. However, the preexisting immunity to chimpanzee adenoviruses among African residents seems to have negative effect on SAdV vaccine [Zhang Q, Seto D. 2015. Chimpanzee Adenovirus Vector Ebola Vaccine--Preliminary Report. N Engl J Med 373:775-776.]  The authors should discuss this kind of vectors in this section.

Response: The suggested information is incorporated (lines 340-343; & mentioned again in section 9 lines 388-390).

Reviewer 2 Report

The manuscript: "Significance of preexisting vector immunity and activation of innate responses for adenoviral vector-based therapy" is well written and of interest for scientific community. The authors make a brief introduction to viral biology and cell entry mechanisms. Then focus on host immune response, that may modulate efficiency of gene transfer and therefore therapy. That is a relevant ant interest topic that is well covered, there are more publications on this topic, but the authors have acknowledged this minor point, and have highlighted on the most relevant points affectig adenoviral therapy. The PI have been working in adenoviral vectors for a long time and I support publication of this review.

Author Response

Reviewer #2

The manuscript: "Significance of preexisting vector immunity and activation of innate responses for adenoviral vector-based therapy" is well written and of interest for scientific community. The authors make a brief introduction to viral biology and cell entry mechanisms. Then focus on host immune response, that may modulate efficiency of gene transfer and therefore therapy. That is a relevant ant interest topic that is well covered, there are more publications on this topic, but the authors have acknowledged this minor point, and have highlighted on the most relevant points affecting adenoviral therapy. The PI have been working in adenoviral vectors for a long time and I support publication of this review.

Response: We appreciate the encouraging comments.